# Clinically-relevant postzygotic mosaicism in parents and children with developmental disorders in trio exome sequencing data

C.F. Wright [ORCID] [1], E. Prigmore[2], D. Rajan[2], J. Handsaker[2], J. McRae[2], J. Kaplanis[2], T.W. Fitzgerald[3], D.R. FitzPatrick[4], H.V. Firth[2,5] & M.E. Hurles[2]

Mosaic genetic variants can have major clinical impact. We systematically analyse trio exome sequence data from 4,293 probands from the DDD Study with severe developmental disorders for pathogenic postzygotic mosaicism (PZM) in the child or a clinically-unaffected parent, and use ultrahigh-depth sequencing to validate candidate mosaic variants. We observe that levels of mosaicism for small genetic variants are usually equivalent in both saliva and blood and ~3% of causative de novo mutations exhibit PZM; this is an important observation, as the sibling recurrence risk is extremely low. We identify parental PZM in 21 trios (0.5% of trios), resulting in a substantially increased sibling recurrence risk in future pregnancies. Together, these forms of mosaicism account for 40 (1%) diagnoses in our cohort. Likely child-PZM mutations occur equally on both parental haplotypes, and the penetrance of detectable mosaic pathogenic variants overall is likely to be less than half that of constitutive variants.

[1] Institute of Biomedicine and Clinical Science, College of Medicine and Health, University of Exeter, RILD Building, Royal Devon and Exeter Hospital, Exeter EX2 5DW, UK. [2] Wellcome Sanger Institute, Wellcome Genome Campus, Hinxton, Cambridge CB10 1SA, UK. [3] European Bioinformatics Institute (EMBL-EBI), Wellcome Genome Campus, Cambridge CB10 1SD, UK. [4] MRC Human Genetics Unit, University of Edinburgh, Edinburgh EH4 2XU, UK. [5] Clinical Genetics, Box 134 Addenbrooke's Hospital, Cambridge University Hospitals, Cambridge, UK. Correspondence and requests for materials should be addressed to C.F.W. (email: caroline.wright@exeter.ac.uk)

Mosaicism is a well-described biological phenomenon in which individuals harbour two or more populations of genetically distinct cells as a result of postzygotic mutation[1,2]. Mutations that occur during early embryonic mitoses can result in somatic and/or germline mosaicism at appreciable levels across multiple tissues[1,2]. However, mosaicism is frequently overlooked as a source of pathogenic variation in rare monogenic diseases largely due to the challenges associated with variant detection[3,4]. Postzygotic de novo mutations (DNMs) may result in somatic mosaicism, potentially causing a less severe and/or variable phenotype compared with the equivalent constitutive mutation, or somatic and gonadal mosaicism, potentially enabling transmission of a pathogenic variant from an unaffected parent to their affected offspring (Fig. 1a)[3–6]. In addition to making an accurate diagnosis in the child, parental mosaicism also has important clinical implications for counselling parents about recurrence risk (Fig. 1b), with substantially increased risk in parental mosaicism but minimal risk in postzygotic mosaicism originating in the proband[7–10]. Numerous cases exist in the literature of recognised monogenic disorders that are occasionally caused by mosaic variants[11], which can range from small sequence variants such as single nucleotide variants (SNVs) and insertion/deletions (indels)[12–16], to large structural variants including copy number variants (CNVs)[17,18] and chromosomal aneuploidy[19–21]. Mosaic variants have also been shown to contribute to the risk of autism spectrum disorders[22,23]. Some pathogenic mosaic variants have been shown to exhibit markedly different abundances in different tissues, due in part to differential negative and positive selective pressures in different tissues[13,24]. This differential tissue representation can be clinically relevant, particularly as some pathogenic mosaic mutations are typically absent from tissues commonly sampled for genetic testing, i.e. blood[13,17]. It is not known how common this phenomenon of differential tissue representation is for mosaic pathogenic sequence variants across different disorders.

Despite its clinical importance, postzygotic mosaicism (PZM) can easily be missed or the variants wrongly assumed to be constitutive due to the technical challenges inherent in the detection of alleles present in only a subset of cells. Next-generation sequencing (NGS) technologies offer an opportunity

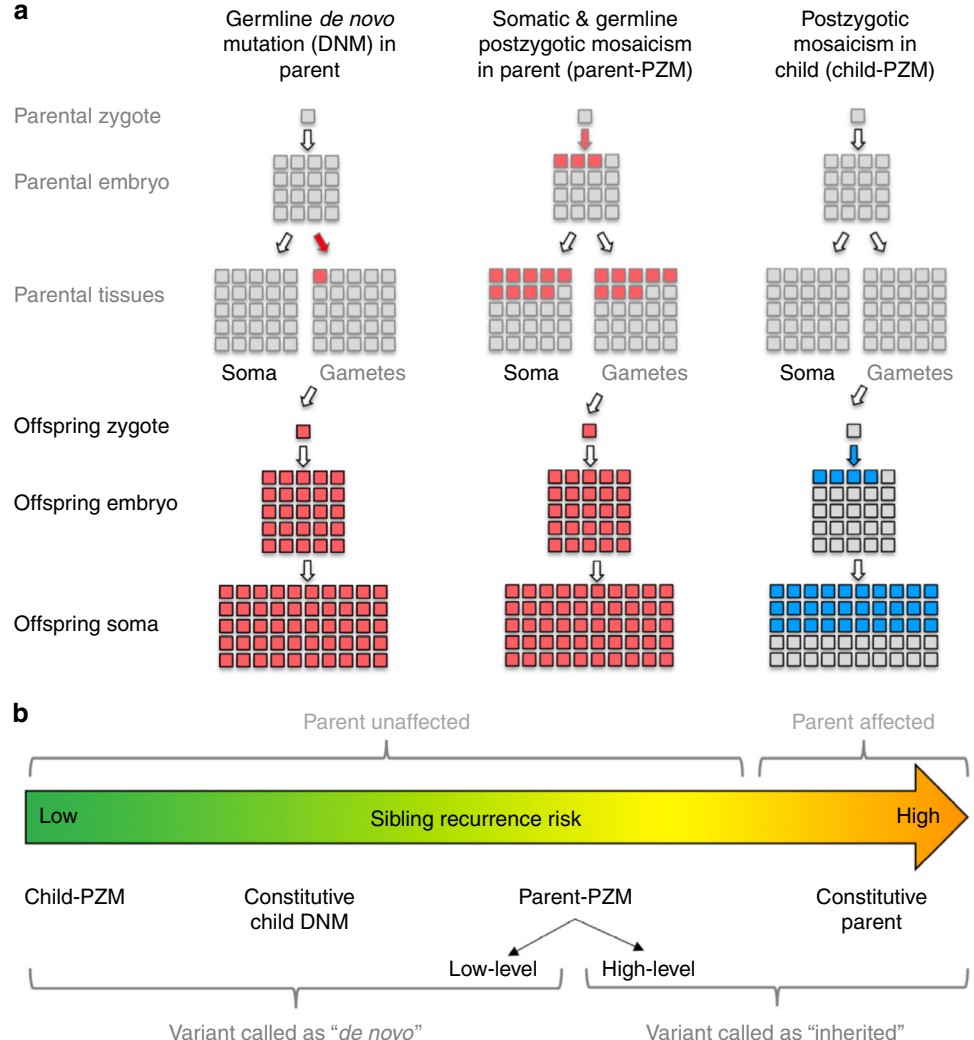

**Fig. 1** Timing of de novo mutations. **a** De novo mutations (DNMs) can occur at any point prior or during development of the embryo, potentially resulting in mosaicism. The most common form of DNM (left) occurs in the parental gametes and is constitutive in the child. A postzygotic (PZM) DNM in the parent (middle) results in mosaicism across multiple parent tissues and a constitutively inherited variant in the child when the parental gametes are affected. A PZM DNM in the child (right) is not present in any parental tissue but is mosaic in the child. **b** Sibling recurrence risk varies with timing of PZM, from very low in child-PZM, to medium in parent-PZM, to high (50%) in affected parents with constitutive pathogenic variants. Different types of mosaic variants can be misclassified by standard variant callers, resulting in erroneous risk estimation or missed diagnoses

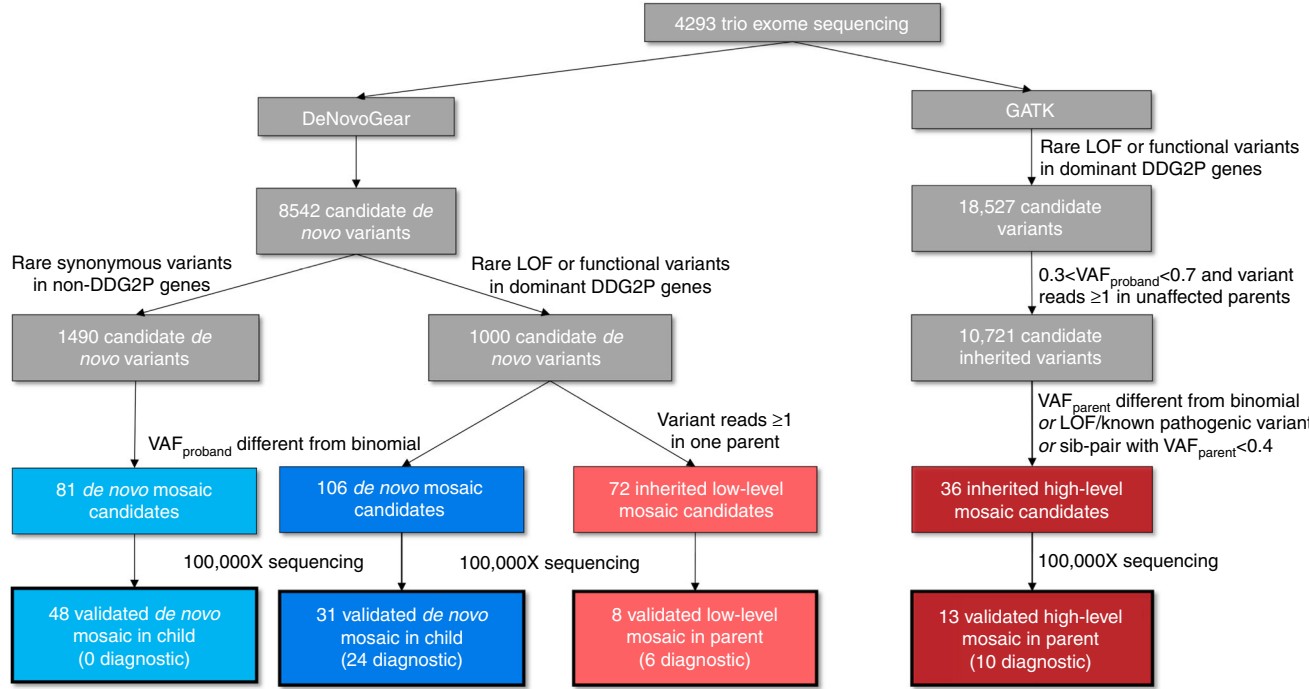

**Fig. 2** Variant selection flowchart. Flowchart outlining variant selection for high-depth sequencing validation experiments. Candidate DNMs from trio exome sequencing were used to select candidate mosaic DNMs in the proband (blue) as well as 'low level' mosaic variants in a parent (light red) for validation; candidate inherited variants were used to select 'high level' mosaic variants in a parent (dark red) for ultrahigh-depth sequencing

to detect lower levels of mosaicism more readily than was previously possible using either capillary sequencing or microarray-based techniques[25–27]. Standard whole-exome sequencing (WES) typically produces data of >30X average depth, allowing detection of alleles only present in a small fraction of reads[28], while ultrahigh-depth (>10,000X) sequencing technologies can be used to detect or confidently confirm mosaic variants present in just a tiny fraction of cells[29]. However, using NGS technologies, mosaic variants present in the majority of cells will often be detected by variant calling tools optimised for constitutive variation and may therefore falsely be assumed to be constitutive. This can lead to under-appreciation of mosaicism in probands and over-estimation of recurrence risks. More importantly, falsely classifying parental PZM as constitutive in both parent and child can lead to missed diagnoses due to the exclusion by bioinformatics pipelines of variants inherited from unaffected parents under the Mendelian assumptions that are typically applied in fully pene-trant conditions.

Large-scale analyses of mosaic pathogenic structural variants has been highly informative for determining the relative contributions of different classes of variants, especially in developmental disorders[18,21]. By contrast, the study of parental and child mosaicism of pathogenic small sequence variants has been relatively piecemeal, focused primarily on specific disorders or sub-types of mosaicism[16,22,25,28,30]. Larger-scale analyses of likely benign sequence variants has been informative about the general properties of post-zygotic mutations[9,10]. A large-scale systematic analysis of mosaic pathogenic sequence variants is needed to determine the relative impact of different classes of mosaicism to disease.

We have previously used trio-WES and SNP-array data from the Deciphering Developmental Disorders (DDD) Study to detect pathogenic mosaic structural variants in children with severe developmental disorders, and found substantial differences in copy number and loss-of -heterozygosity events between blood and saliva (mosaic variants in saliva were often absent from blood)[17,18]. Because of the large burden of pathogenic DNMs in developmental disorders[31], which account for around three-quarters of the total diagnostic yield in this cohort[32,33], we hypothesised that an appreciable number of pathogenic mutations were likely to be mosaic.

Here, we analyse trio-WES data from 4293 parent-offspring families in the DDD Study[34] to find pathogenic PZM—either in the affected child (child-PZM) or an unaffected parent (parent-PZM)—that might be incorrectly annotated or missed by standard NGS pipelines (Fig. 2). We apply lenient thresholds to identify candidate pathogenic PZMs with to ensure high sensitivity and use targeted ultra-deep sequencing to comprehensively validate candidate PZMs. For child-PZM, we observe that the level of mosaicism is usually equivalent in both saliva and blood, and estimate that ~3% of causative de novo mutations exhibit PZM. Likely child-PZM mutations occur in equal proportions on the maternal and paternal haplotypes and, unlike constitutive DNMs, show no evidence of a parental age effect. We observe a marked reduction in the enrichment of damaging, likely patho-genic DNMs in known DD-associated genes with reducing levels of mosaicism.

We also identify parental PZM in 21 trios and overall detect 40 (1%) diagnoses resulting from mosaic variants in our cohort.

## Results

**Postzygotic mosaicism in the child (child-PZM).** To analyse child-PZM, we identified a high-sensitivity set of 8464 candidate DNMs in 4293 children with DDs from trio-WES data using saliva-extracted DNA or blood-extracted DNA in around a third of probands (as previously described[31]). We designated 1000 loss-of-function (LOF) and functional DNMs in autosomal dominant DD-associated genes and X-linked dominant DD-associated genes in females as being likely pathogenic and plausibly causa-tive, and 1490 synonymous DNMs in non-DD-associated genes as being likely benign and unrelated to the DD. To select can-didate mosaic DNMs, we used a variant prioritisation strategy

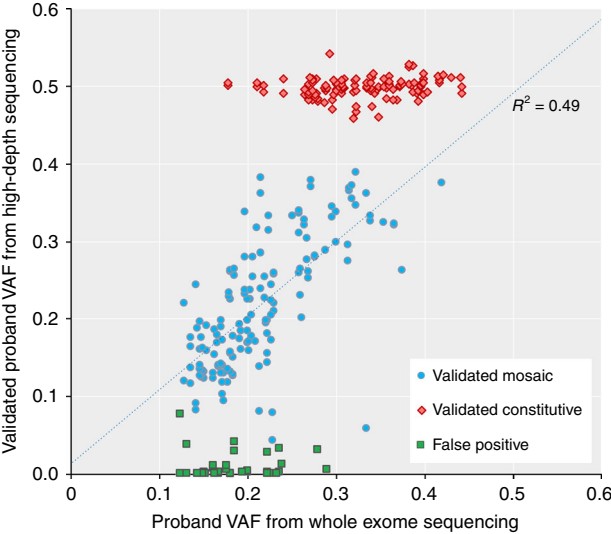

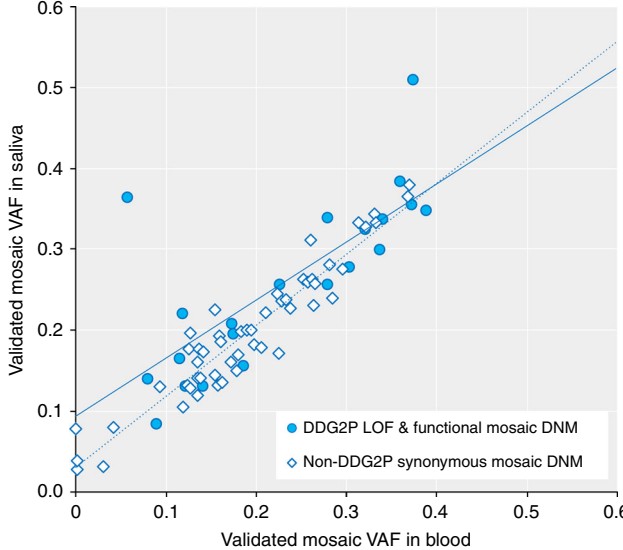

**Fig. 3** Summary of validation results for de novo mutations. Comparison of the variant allele fraction (VAF) from exome sequencing versus from ultrahigh-depth sequencing for candidate DNMs, separated into validation classes: constitutive DNM (red), mosaic DNM (blue) and false positive (green). Validation data from both saliva and blood samples are included where available. The correlation between exome and validation VAFs ($R^2 = 0.49$) includes only mosaic variants

**Fig. 4** Levels of mosaicism in saliva versus blood Comparison of the validated mosaic fraction using ultrahigh-depth sequencing in blood. versus saliva for mosaic DNMs that are either plausibly pathogenic loss-of-function (LOF) or functional variants in known developmental disorder genes (DDG2P, open diamonds), or likely benign synonymous variants in genes not currently associated with developmental disorders (not-DDG2P, filled circles). There is generally a good correlation between saliva and blood VAFs (DDG2P functional, solid line, $R^2 = 0.6$; non-DDG2P synonymous, dotted line, $R^2 = 0.9$) and there is no significant difference between the association of mosaic VAF in saliva versus blood using linear regression and including DDG2P status as a covariate

(see Methods) based on deviation of the proband variant allele fraction (VAF, defined as the number of alternative reads divided by the total read depth) from a binomial distribution centred around 0.5. Using this strategy, we selected 106 likely pathogenic DNMs and 143 likely benign DNMs (of which 81 had both saliva and blood-extracted DNA samples available) for ultrahigh-depth sequencing in all members of the trio. Of the likely pathogenic DNMs: 31/106 validated as mosaic DNMs in the proband (31/1000 or ~3% of likely pathogenic DNMs), of which 20 could be assayed in both blood and saliva samples; 64 validated as constitutive DNMs; three were false positives; and the validation data were uninformative in eight cases (Fig. 3 and Supplementary Fig. 1). Of the likely benign variants: 48/81 validated as mosaic DNMs in both blood and saliva samples (48/1490 or ~3% of benign DNMs); eight validated as constitutive DNMs; 23 were false positives; and the validation data were uninformative in two cases (Fig. 3 and Supplementary Fig. 1).

Validated VAFs of mosaicism in the proband varied from 0.04–0.39 (equivalent to presence of the variant in 8–80% of cells). Similar levels of mosaicism were observed in saliva and blood (Fig. 4) in both likely pathogenic and benign variants, apart from two likely pathogenic variants in *SCN1A* and *SCN8A* that appeared to have significantly higher VAF in saliva than blood. *SCN1A* has previously been shown to harbour differential levels of mosaicism across different tissues[35], and both genes are well-known sources of mosaic diagnoses in epilepsy[30,36,37]. The likelihood that a variant validated as mosaic in the proband was strongly correlated with both the WES variant allele fraction (VAF) and the strength of statistical evidence (binomial *p*-value) for a deviation of VAF away from 0.5 (Table 1 and Supplementary Fig. 1). Twenty-four of the 31 mosaic DNMs were considered to be definitely or likely pathogenic for DD following a detailed clinical evaluation; the remaining seven variants were considered benign either due to a lack of clinical fit ($n = 4$) or the presence of another more plausible genetic diagnosis ($n = 3$). The children with diagnoses in genes that cause well-known syndromes had phenotypes consistent with those syndromes. We were unable to determine whether the

phenotypes were milder as a result of mosaicism, as knowledge of the phenotypic spectrum and individual developmental profiles of the specific disorders is limited and, together with the small numbers of affected patients, constrains drawing definitive conclusions. However, we identified a mosaic (VAF = 0.33) LOF variant in *KMT2D* that was considered fully diagnostic but had an intermediate methylation signature that failed to classify as either benign or pathogenic (Fig. 5).

**Postzygotic mosaicism in the parent (parent-PZM)**. We took two complementary approaches to evaluate parent-PZM. First, to detect low-level parental mosaicism where the variant is not called in the parental sample by standard variant calling algorithms, we started with the same list of 1000 likely pathogenic candidate DNMs described above and selected all variants with one or more alternative allele reads in a single parent for ultrahigh-depth sequencing. Of the 72 candidate variants: eight were validated as constitutive in the proband and mosaic in a parent (1% of the likely pathogenic DNMs), five maternal and three paternal; 53 variants were validated as true DNMs in the proband but absent from the parent; one variant was constitutive in both the parent and proband; six variants were false positives; and the validation data were uninformative in four cases. The number of alternate reads in the parent was highly correlated with the likelihood of validating as parent-PZM: with just a single alt read ($n = 60$), only 5% of sites validated as mosaic, while 42% of sites with two or more reads validated as mosaic in the parent ($n = 12$) (Table 1 and Supplementary Fig. 2). Thus, the majority of candidate sites in which a single read supporting the alternate allele was observed in the parental exome data were due to sequence errors and not mosaicism, and is consistent with the previously estimated sequencing error rate of the Illumina sequencing platform and the depth of exome sequencing in this

**Table 1 Metrics for identifying different types of mosaic variants from whole exome sequencing data. Values are based on ultra-high-depth sequencing validation results, and any uninformative results were excluded**

| Type of mosaicism | Variant caller | Mosaic prioritisation | Validated mosaic | Validated constitutive or FP | Sensitivity | Specificity | PPV | NPV |
|---|---|---|---|---|---|---|---|---|
| De novo (child-PZM) | DNG | Binomial $p < 0.0001$ | 56 | 17 | 69% | 84% | 73% | 80% |
| | | Binomial $p \geq 0.0001$ | 24 | 81 | | | | |
| | | VAF < 0.27 | 63 | 32 | 78% | 68% | 64% | 84% |
| | | VAF ≥ 0.27 | 17 | 66 | | | | |
| Inherited (low-level parent-PZM) | DNG | <2 alt reads in one parent | 3 | 57 | 63% | 89% | 42% | 95% |
| | | ≥2 alt reads in one parent | 5 | 7 | | | | |
| Inherited (high-level parent-PZM) | GATK | Binomial $p < 0.0001$ | 9 | 7 | 69% | 67% | 56% | 78% |
| | | Binomial $p \geq 0.0001$ | 4 | 14 | | | | |
| | | VAF < 0.27 | 13 | 12 | 100% | 43% | 52% | 100% |
| | | VAF ≥ 0.27 | 0 | 9 | | | | |

*PZM* postzygotic mosaicism, *DNG* DeNovoGear, *GATK* Genome Analysis Toolkit, *VAF* variant allele fraction from exome sequencing data, *FP* false positive, *Binomial p* = binomial test on the alternative allele reads, centred around 0.5, *PPV* positive predictive value, *NPV* negative predictive value

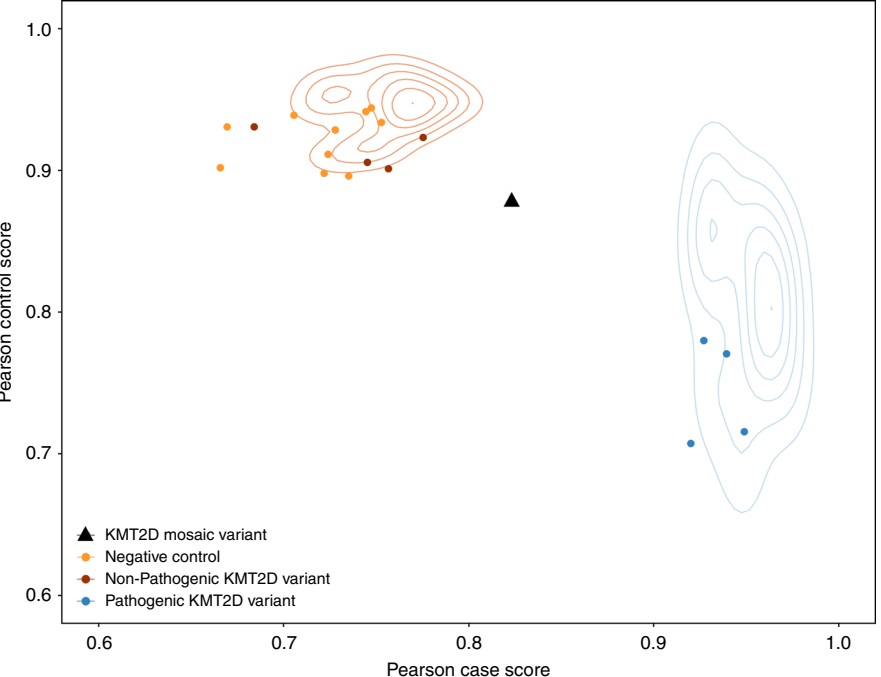

**Fig. 5** Mosaic DDD proband with an intermediate methylation profile. We evaluated DNA methylation at ~850,000 CpG sites across the genome in a subset of individuals from the DDD study using an Illumina EPIC methylation array. The blue cloud represents the distribution of published pathogenic *KMT2D* variants ($n = 12$), and the orange cloud non-pathogenic *KMT2D* variants ($n = 15$) from the published test set[44]. The points in red, orange, and blue represent methylation profiled non-mosaic *KMT2D* cases and negative controls from the DDD study; one DDD proband with a mosaic *KMT2D* stop-gain DNM (black triangle, VAF = 0.33) failed to classify as a case and sits between the two clusters

study[38]. Validated fractions of low-level mosaicism in the parent varied from a VAF of 0.005–0.20 (equivalent to presence of the variant in 1–40% of cells). Six of the eight low-level mosaic inherited variants were considered to be definitely or likely pathogenic following a detailed clinical evaluation; one of these six families had a second sibling with the same disorder emphasising the importance of this analysis for genetic counselling.

Second, to detect high-level parent-PZM where the variant is called in the parental sample by standard variant calling algorithms (and thus might be assumed to have been constitutively inherited), we identified all 18,527 rare LOF and functional variants in dominant DDG2P genes detected in the same 4293

children with DD. We then selected 10,721 constitutive heterozygous variants in the child that were also detected in just one apparently unaffected parent. Given the low prior likelihood of inherited variants being pathogenic in this cohort[33], the vast majority of these are expected to be constitutive in the parent and benign. Using a variant prioritisation strategy based on deviation of the parental VAF from a binomial distribution centred around 0.5, augmented by knowledge of known pathogenic variants, LOF variants and probands with affected siblings (see Methods), we selected 36 candidate parental mosaic variants for validation using ultrahigh-depth sequencing. Of these: 13 variants validated as constitutive in the proband and mosaic in a parent (<0.1% of all inherited rare LoF and functional variants in DDG2P genes),

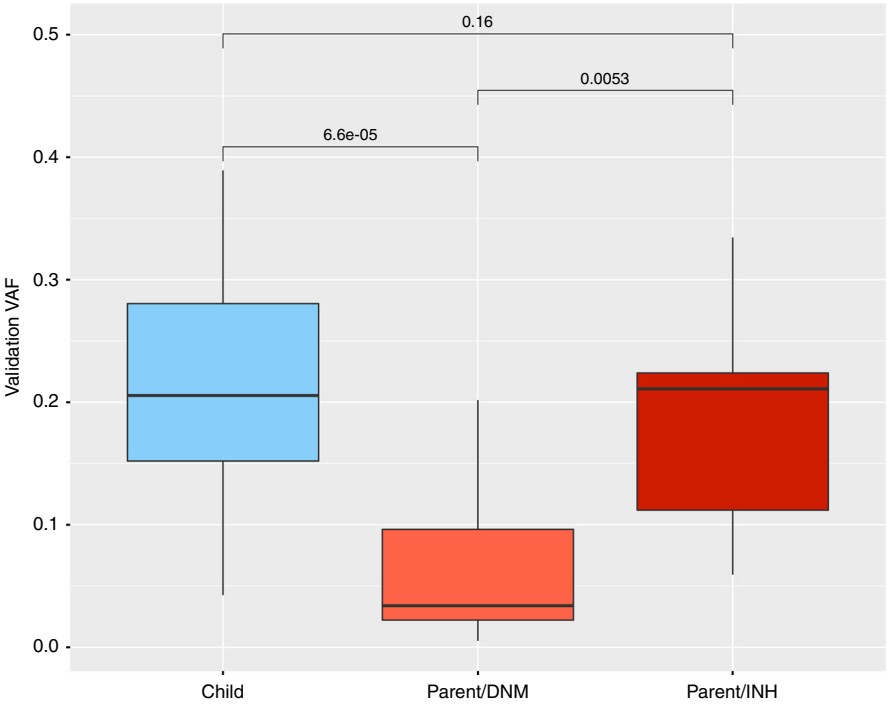

**Fig. 6** Levels of mosaicism across validation experiments Boxplot of estimated level of mosaicism in validated mosaic DNMs, expressed as the variant allele fraction (VAF) from ultrahigh-depth sequencing, in the proband (blue) versus the parents (red). Parental variants are split into two distinctive groups: 'low level' (Parental/DNM, light red), which may be erroneously mistaken for DNMs in the child, and 'high level' (Parental/INH, dark red), which may be erroneously mistaken for constitutively inherited variants. *P*-values calculated using a Mann–Whitney–Wilcoxon test in *R*. Centre line = median; bounds of box = interquartile range; whiskers = quartile ± 1.5 × interquartile range

eight maternal and five paternal; 21 were constitutive in both the parent and proband; and the validation data were uninformative in two cases. The likelihood that a variant validated as mosaic in the parent was correlated with both the parental VAF and the strength of statistical evidence (binomial p-value) for a deviation of VAF away from 0.5 (Table 1). Levels of mosaicism in the parent varied from 0.06–0.33 (equivalent to presence of the variant in 12–66% of cells). Ten of the 13 high-level mosaic inherited variants were considered to be definitely or likely pathogenic for DD following a detailed clinical evaluation; two of these ten families had a second sibling with the same disorder, though they were prioritised for validation on the basis of VAF rather than their presence in affected sib pairs.

**Comparing analytical approaches to identify PZMs**. The diagnostic yield from these combined mosaic analyses was ~1%, and the 40 validated definitely or likely pathogenic mosaic variants are summarised in Supplementary Data 1. The best method for detecting mosaic variants from trio-WES data depends upon the type of mosaicism: for high-level mosaicism (child-PZM or parent-PZM), VAF is both intuitive and sensitive for detecting mosaicism, while the binomial p-value is more specific as it takes into account read depth. Using our exome sequencing trio data, 0.1 < VAF < 0.27 or binomial *p*-value < 0.0001 are good predictors of high-level mosaicism. For low-level mosaicism, where the variant is not detected using standard algorithms and thus we were only able to evaluate it here for parent-PZM, observing two or more reads supporting the variant allele in a parent is a good predictor parent-PZM in putative DNMs (Table 1).

The level of mosaicism detected with ultrahigh-depth sequencing was significantly higher for child-PZM than for low-level parent-PZM ($p = 9.3 \times 10^{-5}$) and higher for high-level than low-level parent-PZM ($p = 0.0053$, Fig. 6). These results support the

intuitively attractive hypothesis that the level of mosaicism is correlated with the likelihood of having a pathogenic impact. To assess this hypothesis more formally, we grouped variants from a high-stringency set of candidate DNMs (defined by setting filtering thresholds to ensure that the number of observed synonymous DNMs equalled the number expected under a null mutation model, i.e. assuming no enrichment for synonymous DNMs in our cohort)[31] into six bins of proband VAF (see Methods). We further subdivided the DNMs into classes of variant consequence (LOF and functional versus synonymous) and types of genes (known DD-associated genes versus all genes, Table 2). The previously observed enrichment[31] of both LOF and functional DNMs in DD-associated genes in our cohort is substantially reduced for mosaic variants ($p = 0.0001$). The correlation between enrichment of potentially damaging variants and increasing VAF is greater in classes of genes where the variants are more likely to cause monogenic developmental disorders (Fig. 7). The linear regression coefficient for genes not currently known to be DD-associated was not significantly different from the null ($p = 0.03$), while the regression coefficients were significantly different for either DD-associated ($p = 0.00007$) or monoallelic DD-associated genes with a loss-of-function mechanism ($p < 0.00001$). For likely mosaic variants (VAF < 0.27) in known DD-associated genes with a monoallelic loss-of-function mechanism, the enrichment of LOF and functional variants is reduced to approximately a third that of constitutive variants ($p = 0.04$).

Using just high-stringency candidate DNMs where it was possible to ascribe the mutation to either the maternal or paternal haplotype ($n = 771$, see Methods), the enrichment of paternal-origin mutations was significantly different ($p = 0.0001$) between mosaic ($n = 41$) and constitutive variants ($n = 730$). We visually inspected IGV plots for all 41 candidate mosaic DNMs with a

**Table 2 Summary of candidate DNMs observed in 4293 DDD trios, sub-divided by variant allele fraction, variant consequence and gene class. Note this table includes all candidate DNMs; it is not limited to those with validation data, and thus includes some likely false positives particularly at lower VAFs**

| VAF bin | Mean VAF | All genes | | | All DDG2P genes | | | MonoLOF DDG2P genes | | | (LOF+Func)/Syn variants | | |
|---|---|---|---|---|---|---|---|---|---|---|---|---|---|
| | | Func | LOF | Syn | Func | LOF | Syn | Func | LOF | Syn | Func | LOF | Syn |
| 0.10-<0.15 | 0.14 | 10 | 1 | 7 | 3 | 0 | 1 | 1 | 0 | 0 | 1.57 | 3.00 | – |
| 0.15-<0.20 | 0.18 | 81 | 18 | 33 | 5 | 3 | 3 | 3 | 2 | 0 | 3.00 | 2.67 | – |
| 0.20-<0.25 | 0.22 | 99 | 35 | 42 | 16 | 8 | 4 | 6 | 4 | 1 | 3.19 | 6.00 | 10.00 |
| 0.25-<0.30 | 0.27 | 135 | 58 | 53 | 23 | 21 | 5 | 7 | 15 | 2 | 3.64 | 8.80 | 11.00 |
| 0.30-<0.35 | 0.33 | 266 | 81 | 69 | 54 | 31 | 10 | 17 | 23 | 3 | 5.03 | 8.50 | 13.33 |
| 0.35-1.00 | 0.49 | 3709 | 1224 | 1084 | 673 | 488 | 112 | 302 | 389 | 28 | 4.55 | 10.37 | 24.68 |
| Total | 0.46 | 4300 | 1417 | 1288 | 774 | 551 | 135 | 336 | 433 | 34 | 4.44 | 9.81 | 22.62 |

*VAF* variant allele fraction from exome sequencing data, *LOF* loss-of-function variants, including splice donor, splice acceptor, stop gained, frameshift and initiator codon variants, *Func* functional variants, including variants include missense, in-frame deletion and in-frame insertion variants, *Syn* synonymous variants, *DDG2P* developmental disorders gene-2-phenotype list (July 2015 version), *MonoLOF* monoallelic and loss-of-function mechanism

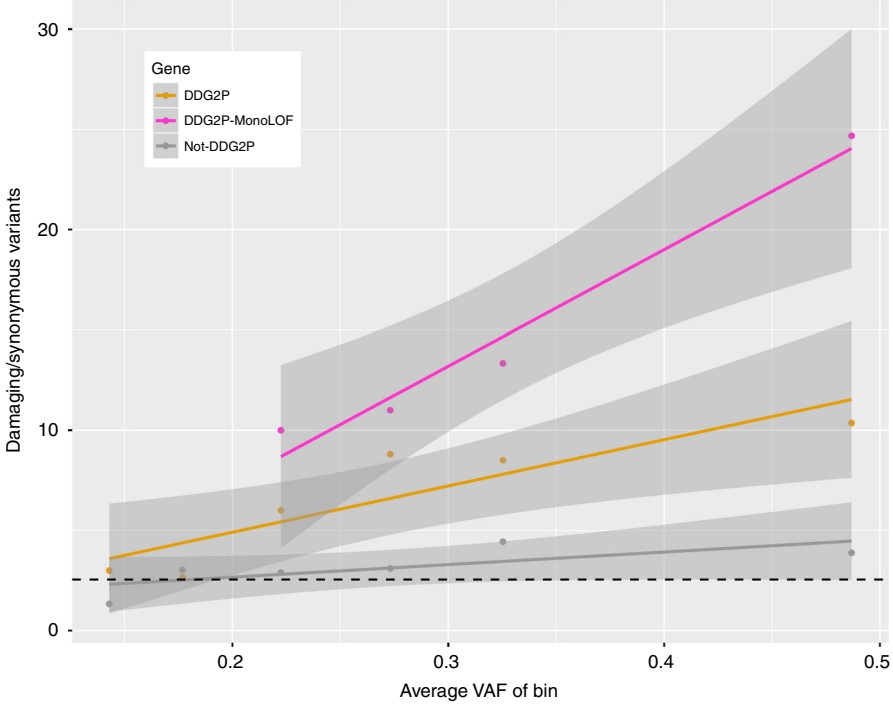

**Fig. 7** Enrichment of damaging variants with level of mosaicism. The exome sequencing variant allele fraction (VAF) for candidate DNMs is plotted against the number of damaging (loss-of-function and functional) variants divided by the number of synonymous variants in each of six VAF bins (0.10– < 0.15, 0.15– < 0.20, 0.20– < 0.25, 0.25– < 0.30, 0.30– < 0.35 and >0.35). The number of variants in each VAF bin was further divided into three subgroups of genes, and the significance of the linear regression coefficient versus the null was assessed using a Z-test: genes not currently known to be DD-associated (Not-DDG2P, gray; $p = 0.03$), genes known to be DD-associated (DDG2P, orange; $p = 0.00007$), and known monoallelic DD-associated genes with a loss-of-function mechanism (DDG2P-MonoLOF, magenta; $p < 0.00001$). Gray zones represent 95% confidence intervals. Null values assuming no enrichment (dotted black line) were calculated from published values of expected numbers of variants in each class based on ExAC, adjusted for indels, and summed across all genes

nearby informative allele inherited from a single parent on the same read-pair. For all validated cases ($n = 6$), we observed the characteristic three-haplotype pattern expected for mosaic variants, and the average VAF of the inherited allele was 0.5; across all 41 variants, there is a significant difference between the VAF of the candidate mosaic DNM and the inherited variant ($p < 2e-10$, Supplementary Fig. 3), supportive of most of the variants being mosaic. Unlike constitutive de novo mutations, which occur more frequently in the paternal gametes and show a strong paternal age effect[31,39], the ratio of candidate mosaic variants on the paternal versus maternal haplotype was 0.95 (21:20), consistent with a 50:50 ratio expected for a postzygotic mutation

arising during early embryo development, before sexual differentiation. Moreover, mosaic variants did not increase appreciably with parental age and the relationship between the number of mosaic DNMs per person and increasing parental age was not significantly different from the null (Supplementary Fig. 4).

## Discussion

We have described an approach for detecting mosaic variants caused by postzygotic DNMs in either a proband or an unaffected parent from trio-WES data. As a result of these analyses, we have made 40 mosaic diagnoses in our cohort of 4293 children with

severe developmental disorders, of which 24 are child-PZM and 16 are inherited parent-PZM (Supplementary Data 1). The clinical implication is substantially different between these two groups: the recurrence risk in future siblings of children with child-PZM is negligible, while that of children with inherited parent-PZM is perhaps as high as 50% depending upon the level of parental mosaicism[7–10], and indeed three of our 16 probands diagnosed with inherited parental mosaic variants have siblings with the same disorder.

In principle, diagnoses of DNMs could be split into three groups based on risk of recurrence in future siblings: moderate (parental somatic mosaicism, ~1–2% of DNMs), low (no parental somatic mosaicism but possible parental gonadal mosaicism, ~95% of DNMs) and minimal (child mosaic, ~3% of DNMs). The overlapping VAF distributions of PZMs in affected probands and apparently unaffected parents (Fig. 6) indicates that different levels of mosaicism may be necessary for pathogenicity in different genes or developmental phenotypes[40,41], and highlights the importance of considering variants called in parents as well as apparent DNMs to capture the full range of parental mosaicism. We estimate that, in our cohort, likely pathogenic mosaic DNMs are enriched in DD-associated genes at around half the level observed for constitutive DNMs (Fig. 7). While the penetrance of pathogenic DNMs undoubtedly increases with increasing VAF, pathogenic PZMs are still likely associated with reduced reproductive fitness. This suggests that the VAF distribution of pathogenic PZMs in unaffected parents is likely to be depleted of higher VAFs compared to benign PZMs, which brings into question the accuracy of recurrence risk calculators based on benign PZMs[9]. Importantly, we observe that, unlike constitutive DNMs, likely child-PZM shows no parental bias or evidence of a parental age effect.

The clinical relevance of detecting mosaicism is not limited to counselling of recurrence risks. An increasing number of developmental disorders also have confirmatory biomarker assays whereby informative molecular signatures help to distinguish pathogenic and benign variants and classify variants of uncertain significance[42–44]. Detecting mosaicism is important for accurate interpretation of confirmatory biomarker assays, wherein a mosaic individual may exhibit an intermediate result that may get mis-interpreted. To exemplify this, we observed a DDD participant with a validated diagnostic child-PZM LOF variant in *KMT2D* who exhibited an intermediate biomarker phenotype for a published DNA methylation signature for Kabuki syndrome caused by mutations in *KMT2D*[44] (Fig. 5). Caution should therefore be used when evaluating the likely pathogenicity of mosaic variants using molecular biomarkers.

Our finding that ~3% of DNMs are mosaic is lower than in some previous studies[3,6,22,45], most likely due to differences in clinical ascertainment, both in terms of phenotype and family history, and different sensitivity for very low level mosaicism (e.g. VAF < 0.1). Detecting likely mosaic variants depends on the depth of the original sequencing data, the error rate of the sequencing platform, and the metrics used to distinguish different types of variants. In principle, it should also possible to estimate the proportion of DNMs that are child-PZMs using mixture modelling on the VAF distribution of DNMs as compared to constitutive variants (Supplementary Fig. 5a), obviating the need for extensive validation. We found that fitting a Gaussian mixture model to these distributions gives an estimate of 6% of DNMs that are potentially mosaic (at VAF > 0.1), with mean VAF = 0.198, compared with 0% of the inherited variants (Supplementary Fig. 5b). This is likely to be an over-estimate compared with the 3% estimate from our ultrahigh-depth validation experiments, where around half the candidate variants validated as mosaic, suggesting that the presence of false positives and constitutive variants confounds this kind of analysis.

An obvious limitation of our approach was the depth of sequencing (~50X) and, as a consequence, only being able to detect mosaicism where the fraction was appreciably greater than the sequencing error rate. Since DeNovoGear requires >2 ALT reads to detect a variant, we estimate our power to detect child-PZM was ~75% at VAF = 0.1 and ~90% for VAF = 0.2–0.3 (see Supplementary Fig. 6). We used a high-sensitivity FDR-based approach to prioritise candidate mosaic variants, and preliminary empirical analyses suggested that detecting candidate de novo mosaic variants with a VAF of <0.1 using this strategy would be overwhelmed with false positives, so we introduced this VAF threshold to focus on a more tractable VAF range. Although mosaic mutations present in >20% of cells are more likely to have occurred in early embryogenesis, and thus have a syndromic developmental impact, future work is needed to investigate low-level child-PZMs (VAF < 0.1). These limitations are likely to be common to many existing diagnostic pipelines but may be reduced by the use of amplification-free whole genome sequencing, improved modelling of sequencing errors, high-depth targeted sequencing or long-read sequencing, which would also improve variant phasing. Validation of very low levels of mosaicism might also be improved by using unique molecular indexes[46,47].

We only had access to DNA extracted from two tissues (saliva and blood for probands and only saliva for parents) in which to explore tissue specificity. The correlation between the level of mosaicism in these tissues versus developmentally-relevant organs, such as the brain, is unknown and thus caution is needed when considering the correlation between observed VAF and phenotype or recurrence risk. Indeed, paternal germline mosaicism might be better evaluated using a paternal sperm sample[8,48]. Nonetheless, unlike our previous findings on mosaic structural variants[17,18], we did not observe any difference between blood and saliva samples in the child-PZMs, as might be expected for PZMs that arise very early in embryogenesis. Although inter-individual variability and cellular heterogeneity make direct saliva-blood comparisons challenging[49], our results suggest that concerns about failing to detect pathogenic mosaic variants by testing blood rather than saliva may be less important for sequence variants than for structural variants.

In conclusion, using 4293 families with severe developmental disorders, we have shown that mosaic variants are a significant cause of rare developmental disorders, and that they can be detected from standard trio WES. The relatively low proportion of mosaic variants limited our ability to explore decreasing pathogenicity with decreasing VAF in a more quantitative manner, as well as evaluate recurrence risk empirically based on affected siblings, for which much larger cohorts will be needed. Finally, we did not have sufficient numbers of variants in any given gene to assess whether mosaic DNMs in the proband reduced the severity or expressivity of the child's developmental phenotype relative to constitutive DNMs. There is now a need for very large-scale studies to address remaining questions and further refine our current findings regarding the relationship between variant pathogenicity and mosaicism level, further investigate tissue specificity (including gametes) and empirically estimate the recurrence risk from siblings. To empower this research endeavour, as well as increase diagnostic yields and improve clinical management, algorithms tuned to detecting mosaic pathogenic variants—particularly in an unaffected parent —should be integrated into paediatric genomics pipelines.

## Methods

**Patient recruitment and data collection**. The DDD Study has UK Research Ethics Committee approval (10/H0305/83, granted by the Cambridge South REC, and GEN/284/12 granted by the Republic of Ireland REC). Patients with severe, undiagnosed developmental disorders and their parents were recruited and systematically phenotyped by the 24 Regional Genetics Services within the United

Kingdom (UK) National Health Service and the Republic of Ireland. Clinical data (growth measurements, family history, developmental milestones, etc.) were collected using a standard restricted-term questionnaire within DECIPHER[50], and detailed developmental phenotypes for the individuals were entered by the patient's clinician using Human Phenotype Ontology (HPO) terms[51]. Saliva samples for the family trio as well as blood-extracted DNA samples for the probands were collected and processed as described previously[52]. Trio exome sequencing was performed on either saliva or blood using Illumina HiSeq (75-base paired-end sequencing) with SureSelect baits (Agilent Human All-Exon V3 Plus and V5 Plus with custom ELID C0338371). Mapping of short-read sequences for each sequencing lanelet was carried out by the Wellcome Sanger Institute's Human Genetics Informatics team using the Burrows-Wheeler aligner (BWA; version 0.59)[53] backtrack algorithm with the GRCh37 1000 Genomes Project phase 2 reference (also known as hs37d5). Sample- level BAM improvement was carried out using the Genome Analysis Toolkit (GATK; version 3.1.1)[54] and SAMtools (version 0.1.19)[53]. Average read depth in the coding regions across these samples was 50X. Single nucleotide variants and indels were called using the GATK HaplotypeCaller (version 3.2.2) and GATK resource bundle (version 2.2)[55], run in multi-sample calling mode using the complete data set. We used DeNovoGear (DNG, version 0.54)[56] to detect likely DNMs from trio exome BAM files. Variants were annotated with minor allele frequencies (from 1000 Genomes Project[57], the UK10K cohort[58], the Exome Aggregation Consortium[59], and internal data from unaffected parents in the DDD Study) and the predicted consequence (using Ensembl Variant Effect Predictor[49]). The data are available under managed access from the European Genome-phenome Archive (Study ID EGAS00001000775), and likely diagnostic variants are available open access in DECIPHER[51].

**Variant filtering and selection**. We generated a high-sensitivity set of 8542 rare (MAF < 0.01) candidate DNMs from 4293 WES trios. To assess the burden of mosaic SNVs and indels, we calculated the expected number of DNMs as described previously[58,60] based on gene-specific mutation rates that account for gene length and sequence context[31], and increased the stringency of called DNMs until the number of observed synonymous variants equated that expected under the null-mutation model, as published previously[31]. Using this high-stringency set of DNMs, we calculated the proband variant allele fraction (VAF)—defined as the read depth of the alternative allele divided by the total read depth—and calculated the binomial probability of the observed VAF given an expected VAF of 0.5 (i.e. a constitutive heterozygous variant). When identifying candidate DNMs, we applied a filtering threshold on proband VAF > 0.1 due to the difficulty of distinguishing real variants from sequencing errors at lower VAF in WES data of ~50X average depth. As a consequence, all candidate DNMs in this dataset had a proband VAF > 0.1.

To determine the parental origin of child-PZMs, we used informative SNVs inherited from a single parent on the same read-pair as the candidate DNM to determine whether the variant occurred on the maternal or paternal haplotype. To assess how variant pathogenicity varies with the level of mosaicism, we also calculated the number of different classes of DNMs in six bins of increasing proband VAF (0.10– < 0.15, 0.15– < 0.20, 0.20– < 0.25, 0.25– < 0.30, 0.30– < 0.35, > 0.35), excluding variants that fell below our stringency threshold or were on the X or Y chromosomes in males. We annotated variants with predicted loss-of-function (LOF; splice donor, splice acceptor, stop gained, frameshift, initiator codon) or functional consequences (functional; missense, inframe deletion, inframe insertion) in dominant genes known to cause developmental disorders (DDG2P, July 2015 version; www.ebi.ac.uk/gene2phenotype). We then calculated the ratio of potentially damaging (LOF and functional) DNMs to synonymous DNMs in each VAF bin and compared the linear regression coefficients for different classes of genes.

Candidate PZMs were selected for validation by ultrahigh-depth sequencing based primarily on statistical deviation from a VAF of 0.5. To limit the dataset to potential diploid mosaicism, we restricted our analysis of DDG2P genes to those with autosomal dominant inheritance and X-linked dominant inheritance in female probands, and to enrich our dataset for the most likely pathogenic LOF variants, further focused on genes known to cause disease via a LOF mechanism. Variants were then analysed under the following models:

(1) Apparent DNMs: we analysed our high-sensitivity set of 8,464 candidate DNMs (8,542 candidate DNMs excluding 78 X-chromosome variants in boys) under two scenarios:

  a. Candidate child-PZM: a binomial p-value for the proband VAF was calculated assuming a mean of 0.5 then two classes of variants selected for validation:

    i. Likely pathogenic DNMs: LOF and functional variants in dominant DDG2P genes ($n = 1,000$) with a false discovery rate (using binomial $p$ value) of <0.2 ($n = 106$).

    ii. Control variants: synonymous variants in non-DDG2P genes ($n = 1,490$) with a false discovery rate of <0.05 ($n = 143$).

  b. Candidate low-level parent-PZM: candidate likely pathogenic DNMs (see above) with any alternative allele reads in one parent were selected for validation ($n = 72$).

(2) Apparent inherited variants—candidate high-level parent-PZM: we identified 18,527 rare (MAF < 0.001) inherited heterozygous LOF and functional variants in dominant DDG2P genes in 4293 probands, and annotated them with reference and alternative allele read depths across all members of the trio. We excluded variants in children with affected parents, with reads supporting the mutant allele in both parents, or with proband VAF > 0.7 or <0.3, leaving 10,721 candidates. To maximise the detection of clinically relevant parent-PZMs, we adopted three different strategies to identify 36 candidate parent-PZMs for validation from this long list of possible candidates:

  a. The binomial $p$-value for parental VAF was calculated assuming a mean of 0.5; variants were then selected for validation using a false discovery rate (Benjamini-Hochberg) of <0.1 ($n = 20$);

  b. We identified a subset of variants with a high likelihood of pathogenicity (protein-truncating variants and known pathogenic variants) and parental VAF < 0.4 ($n = 10$);

  c. We identified variants present in affected sib pairs with parental VAF < 0.4 ($n = 6$).

**Experimental validation**. Validation of PZM candidates was performed using PCR amplification followed by Illumina MiSeq (250 bp paired-end reads) in all members of the trio, with 40 ng of genomic DNA as template and primers designed to amplify 150–250 bp products centred around site of interest. Median depth of coverage across all validations was 100,032 reads. Both saliva-extracted and blood-extracted DNA samples from the proband were assayed where available; only saliva-extracted DNA was available for parents. Variant inheritance was classified automatically using an in-house pipeline previously described[61] and manually confirmed in IGV[44]. Variant pathogenicity was assessed, for each variant, by at least two consultant clinical geneticists through a composite approach of patient assessment, variant evaluation, inheritance and clinical fit compared with previously published cases of children with pathogenic (usually constitutive) variants in the same gene. A full list of validated mosaic variants is provided in Supplementary Data 2.

To investigate the effect of mosaic mutations on a clinically relevant biomarker, we evaluated DNA methylation at ~850,000 CpG sites across the genome in a subset of individuals from the DDD study using an Illumina EPIC methylation array. A DNA methylation signature that discriminates between benign and pathogenic KMT2D variants was derived as described previously[44] using published data (GEO: GSE97362). CpG sites with at least 10% differential methylation between KMT2D and control samples in the training set were identified and restricted to an FDR < 0.01 using a Mann-Whitney U test at each probe; the resulting CpG sites ($n = 112$) form the methylation signature for KMT2D. We applied this signature to 29 DDD probands profiled on the EPIC array, nine of whom had KMT2D de novo mutations of uncertain pathogenicity and 20 were age and sex matched negative controls. We assessed DNA methylation status at signature CpG sites using the Pearson correlation approach described previously[44] using all 29 DDD probands and an additional 27 KMT2D variants of uncertain pathogenicity from the published test set[44].

## Data availability

All diagnostic variants linked to phenotypes are available via the DECIPHER database (https://decipher.sanger.ac.uk/). DDD Study data is available under managed access via the European Genome-phenome Archive (https://ega-archive.org/studies/EGAS00001000775).

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

## Acknowledgements

We wish to thank everyone in the DDD Study, including the patients and families involved as well as the NHS regional genetics services that recruited them to the study and provide ongoing clinical care. We particularly wish to thank the following clinical geneticists: Alex Henderson, Alex Magee, Amanda Collins, Anand Saggar, Anne Lampe, Astrid Weber, Birgitta Bernhard, Charles Shaw-Smith, David Goudie, Dian Donnai, Diana Baralle, Esther Kinning, Fiona Stewart, Frances Elmslie, Francis Sansbury, Jonathan Berg, Kate Chandler, Katherine Lachlan, Katrina Prescott, Laura Yates, Lily Islam, Meriel McEntagart, Michael Parker, Moira Blyth, Natalie Canham, Nora Shannon, Pradeep Vasudevan, Richard Fisher, Rita Ibitoye, Ruth Newbury-Ecob, Sally Davies, Sarah Smithson, Shane McKee, Susan Tomkins, Tabib Dabir and Tessa Homfray. We also thank Patrick Short for statistical advice, and the scientists at the Wellcome Sanger Institute involved in sample management, DNA sequencing and data processing. The DDD study presents independent research commissioned by the Health Innovation Challenge Fund [grant number HICF-1009–003], a parallel funding partnership between the Wellcome and the Department of Health, and the Wellcome Sanger Institute [grant number WT098051]. DRF is funded by a MRC University Unit for the MRC Human Genetics Unit to the University of Edinburgh. The views expressed in this publication are those of the author(s) and not necessarily those of the Wellcome or the Department of Health. The study has UK Research Ethics Committee approval (10/H0305/83, granted by the Cambridge South REC, and GEN/284/12 granted by the Republic of Ireland REC). The research team acknowledges the support of the National Institute for Health Research, through the Comprehensive Clinical

Research Network. This study uses DECIPHER (https://decipher.sanger.ac.uk), which is funded by Wellcome.

## Author contributions

C.F.W. drafted and finalised the paper; C.F.W. and M.E.H. designed the experiments and analysed the results; E.P. and D.R. performed and analysed the validation experiments; J.H. designed and analysed the methylation experiments; J.M., J.K. and T.W.F. provided bioinformatics support and variant datasets; D.R.F. and H.V.F. provided clinical input for variant interpretation; the D.D.D. Study provided all the infrastructure and resources required for patient recruitment and phenotyping, sample preparation and sequencing, as well as data generation and management; all authors approved the final draft.

## Additional information

**Competing interests:** M.E.H. is a co-founder, consultant and non-executive director of Congenica Ltd. The remaining authors declare no competing interests.

