## [Peer Review File · Nature Communications]

Editorial note: This manuscript has been previously reviewed at another journal that is not operating a transparent peer review scheme. This document only contains reviewer comments and rebuttal letters for versions considered at Nature Communications. Mentions of the other journal have been redacted.

Reviewers' Comments:

Reviewer #2:

Remarks to the Author:

Wright et al. confirm that post-zygotic mutations (PZMs) are a relatively common phenomenon, and show that up to 1% of patients with DD can be explained due to PZMs. The findings may have important consequences for the counselling of respective families. This is a large-scale and systematic study, which is valuable to the field.

The authors have revised their manuscript and responded to most criticism of the reviewer in an appropriate fashion. I propose to now accept this manuscript with minor revisions.

Minor points:

- Rebuttal letter:

o Point 3: as there is no better paper please cite the Thiede et al paper on saliva cell constitution; add a statement on the difficulty of saliva/blood comparison (and maybe inter-individual variability?) and discuss in comparison to the authors own CNV observation. Or show/cite "other groups" saliva work.

o Point 5: "we don't believe that company-specific errors are an appreciable cofounder". This is not about the authors "believe" it's about what they can show. The 41 candidate mosaic sites for which a phase could be determined: how often was a "third allele observed"? This is not yet described in the manuscript now as an "independent way of confirmation for mosaic status" – which they should. Of those: how many of those are amongst the 31 variants shown in the article?

o Point 8: The authors misunderstood this reviewer's comment. PZMs may follow another model (not the standard discovery for DNMs). For some genes already two PZMs may be significant; this may be totally novel genes (e.g. otherwise lethal for germline) and never appear in any germline DNM list. Such an analysis would have boosted the impact of the paper significantly! Have the authors gone back to the original data, and looked into any essential gene lists to see whether any of those genes have multiple candidate PZMs?

- Points from the manuscript:

o Abstract and discussion: The penetrance estimate of PZMs compared to DNMs needs to be addressed with a bit more care. The argumentation for this is not easy to follow. While this observation may be true for the overall list of PZMs; there will be a lot of PZMs that individually show 100% penetrance just like DNMs, so therefore this should be stated more carefully to avoid any misunderstanding.

o Page 7, line 199: As part of the result section it would be good for the reader to understand that the 8464 DNMs are the exact same calls from the original DDD Nature manuscript (cite here accordingly).

o Page 7, line 194: describe 'clinical fit'. What is the expected outcome for a PZM in a gene for which germline DNMs cause severe DD – what about isolated ID.

o Page 8, line 225: Of the 31 proven PZMs only 24 are considered (likely) pathogenic; although all candidate PZMs were considered likely pathogenic. It would be most interesting for the reader to understand the interpretation details for the remaining 7 variants. Did the VAF level influence this? Or what does this say about the initial criteria? Add also to Table 2.

o Page 9, line 288: It remains difficult to understand to what does 'these results' refer. The comparison of parental and patient PZMs may lead to too strong conclusions as these are two separate sets of mutations.

♣ Parental-PZMs are mutations which in germline cause DD (as seen in the respective children); so yes indeed low level in blood of the parents does not cause an obvious phenotype. However were the parents really phenotyped in depths? No subtle phenotype? What about a low level mosaic in brain or other potentially affected tissues, was that excluded?

♣ Patient PZMs: To a large extent the very same mutation was not observed in germline before. Hence viability and penetrance for those isn't fully known. AND: blood VAF may not say too much about e.g. brain VAF.

♣ In summary: The direct correlation between VAF and phenotype in blood seems oversimplified and needs to be discussed with greater care. Revertant mosaicism is not touched at all, which may

be more common in blood.

o Page 11, line 336: Explain 'enrichment' in this context, this is enrichment over what?

o Page 12, line 387: The statement 'missing pathogenic mosaic variants from blood may be considerably less important' is oversimplifying and may be misleading. This may give clinicians the wrong impression that a 2nd sample may not be valuable. There are ample examples that a 2nd (non-blood) sample is crucial e.g. Cornelia de Lange Syndrome, or any other 'organ-specific' or 'organ-restricted' phenotype. This needs to be re-written.

Reviewer #4:

Remarks to the Author:

The authors have sufficiently addressed the concerns raised by Reviewer #1 in the previous round of review. The new analysis they have added, showing an intermediate methylation profile in a mosaic DDD proband, is quite interesting and helps improve the novelty of the study. Although this result only pertains to a single proband, I believe it is an important finding because it directly supports the intuition that mosaicism can lead to milder phenotypes in developmental disorders, as mentioned in lines 46 and 228-230.

Major comments:

1. The findings of the intermediate methylation profile in a mosaic DDD proband are currently buried in the Discussion (lines 344-352) and the methods leading to this result are only superficially mentioned in the caption of Supplementary Fig. 6. These should be moved to the Results and Methods, respectively, and the authors should explain in greater detail the data and methods used for this analysis.

Minor comments:

2. The authors' justification for their use of different FDR thresholds is reasonable, and they appeal to the results shown in Fig. 4 to support their claim that there is no systematic difference in VAF between true mosaic benign mutations and true mosaic pathogenic mutations. This claim could be strengthened by testing the association of mosaic VAF in saliva and mosaic VAF in blood using linear regression and including DDG2P status as a covariate. If the authors' claim is correct, DDG2P status should not significantly improve the model fit. It would also be useful if Fig. 4 showed separate lines-of-best fit for the mosaic benign and mosaic pathogenic data.

Reviewer #5:

Remarks to the Author:

Review of ms 196604 Nat Comms entitled "Clinically-relevant postzygotic mosaicism in parents and children with developmental disorders in trio exome sequencing data" by Wright et al.

This manuscript describes an approach to analyse mosaic mutations (i.e. SNVs and indels – deviation from 50:50 for child-PZM and down to VAF levels ~1% for parent-PZM) in the large WES trio sequencing dataset generated by the DDD consortium (4293 trios).

Although the general approach is not novel and the results are in line with those described in previous studies, overall, I believe that this well-written manuscript represents a comprehensive and helpful addition to the recent body of work on mosaicism. The main attraction of the present study is the systematic approach for identification of mosaic SNVs (both parental- and child-PZM) across a large cohort – although with an average of 50X coverage for WES, the dataset is likely underpowered to identify mosaic cases (and overall only a small number of mosaic mutations are called).

I have a few comments and have identified some relatively minor issues that will need to be

addressed:

1. Line 127 describes “a high-sensitivity set of 8,542 rare (MAF<0.01) candidate DNMs from 4,293 WES trios” but on line 199 (and line 160), the “high-sensitivity set of 8,464 candidate DNMs in 4,293 children with DDs” – please clarify.

2. Despite applying a filtering threshold on proband of VAF>0.1 (line 136), some child-PZM were called and validated at VAF 0.04 (line 217 and Figure 3). This probably relates to Read-Depth but please explain or check phrasing?

3. Some comments on Table 2:

Please indicate the tissue that was used for the WES (relates to the “Mosaic VAF” column)

Please add the read depth for the WES VAF data (although read depth for the cohort is ~50X average, the likelihood to detect mosaic variants depends on local coverage).

The points above are important because the VAFs measured by WES data and the tissue deep-sequencing do not seem to always correlate – could you please plot and comment.

Please add the encoded protein change of the validated pathogenic changes

Patient 273554 with NFIX frameshift mutation – as annotated this patient is a Proband-PZM (VAF ~0.35 in both blood and saliva) but has an affected sib? This makes no sense?

4. I could not find a list of the 49 validated synonymous mutations (nor the other 12 validated mutations that did not make it on the Table 2 of pathogenic mutations) or whether validation was performed in blood and saliva (where available). These data would represent another estimate of the VAF correlation in these 2 tissues and with those measured by WES (I appreciate they are likely to have been plotted on Figure 4 but a supplementary table would be useful).

5. It would be of interest to try to characterise the mutational signature of these mosaic mutations – although I suspect there are not enough calls to perform this analysis?

6. As mentioned in the text for pathogenic mutations, it is possible that there is a tissue-specific enrichment/depletion of mutant cells and therefore caution should be taken in interpreting the VAF levels measured in blood and/or saliva and those that may be present in other organs (including brain or gonads) and hence the correlation between observed VAF and phenotype or recurrence risk – it would be helpful to discuss this point in further details for example on lines 228-230 or 383-387. For some disorders such as Dravet or Cornelia de Lange, this is very relevant.

Related to this, 0.5% of parental-PZM were called – would you please comment on /discuss the benefit to assess germline tissue (i.e. sperm) to properly estimate paternal gonadal mosaicism – this matters also for interpretation of Fig 1B where “sibling recurrence risk” is correlated to low/high-level of Parent-PZM).

Several studies have now shown the power of sperm sequencing for direct recurrence risk estimation for paternally-derived DNMs (see for example Ref 8 - Breuss et al). Of note, this is particularly relevant for some DD genes such as SCN1A (see Yang et al Sci Rep 2017 - doi: 10.1038/s41598-017-15814-7) or MecP2 (doi: 10.1038/s41436-018-0348-2).

7. Among all the validated mosaic (3% of causative child-PZM and the 0.5% parent-PZM), could you indicate how many (and which one, on Table 2 for example) had been previously identified (either as DNM or mosaic) in previously published DDD studies. The abstract (line 31) and text (Line 275) state that together child-PZM and parent-PZM represent an additional 40 diagnoses. Is this so or is it a ‘differential’ diagnosis in some cases.

Sup fig2 – correct “constitutive DNM in child”

Sup fig 3 – please indicate the number of trios for which avge mutations are plotted for each age group.

Clinically-relevant postzygotic mosaicism in parents and children identified from trio exome sequencing data: RESPONSES TO REVIEWERS COMMENTS, Nature Communications

We thank the reviewers for their helpful comments, which we have fully addressed in the revised manuscript and point-by-point below.

Reviewer #2 (Remarks to the Author):

Wright et al. confirm that post-zygotic mutations (PZMs) are a relatively common phenomenon and show that up to 1% of patients with DD can be explained due to PZMs. The findings may have important consequences for the counselling of respective families. This is a large-scale and systematic study, which is valuable to the field.

The authors have revised their manuscript and responded to most criticism of the reviewer in an appropriate fashion. I propose to now accept this manuscript with minor revisions.

Minor points:

- Rebuttal letter:

o Point 3: as there is no better paper please cite the Thiede et al paper on saliva cell constitution; add a statement on the difficulty of saliva/blood comparison (and maybe inter-individual variability?) and discuss in comparison to the authors own CNV observation. Or show/cite “other groups” saliva work. We have added this reference as well as another more recent reference (Theda “*Quantitation of the cellular content of saliva and buccal swab samples*” 2018), and a comment to the discussion about the difference with our own previous observations with mosaic CNVs.

o Point 5: “we don’t believe that company-specific errors are an appreciable cofounder”. This is not about the authors “believe” it’s about what they can show. The 41 candidate mosaic sites for which a phase could be determined: how often was a “third allele observed”? This is not yet described in the manuscript now as an “independent way of confirmation for mosaic status” – which they should. Of those: how many of those are amongst the 31 variants shown in the article?

We thank the reviewer for this suggestion and have visually inspected IGV plots for all 41 candidate mosaic DNMs with a nearby informative allele inherited from a single parent. For all validated cases (n=6), we observed the characteristic three-haplotype pattern expected for mosaic variants, and the average VAF of the inherited allele was 0.5; across all variants, there is a significant difference between the VAF of the candidate mosaic DNM and the inherited variant, supportive of most of the candidate variants being mosaic. The deep sequencing of the non-transmitting parent at the same site provides an estimate of the base-specific error rate due to the sequencing technology and the sequence context, and each mosaic variant is represented in a highly statistically significantly elevated number of reads relative to that error rate. We have added a sentence about this to the manuscript as well as a Supplementary figure.

o Point 8: The authors misunderstood this reviewer’s comment. PZMs may follow another model (not the standard discovery for DNMs). For some genes already two PZMs may be significant; this may be totally novel genes (e.g. otherwise lethal for germline) and never appear in any germline DNM list. Such an analysis would have boosted the impact of the paper significantly! Have the authors gone back to the original data, and looked into any essential gene lists to see whether any of those genes have multiple candidate PZMs?

We thank the reviewer for clarifying this point. We have now gone back to our list of stringent DNMs that are putatively mosaic and evaluated the number of candidate mosaic damaging variants in non-DDG2P but high-pLI genes. We considered high pLI genes to be a more likely source of potential dominant embryonic lethal genes than essential gene lists as many of the latter only have a recessive phenotype. There are just three genes with two candidate functional mosaic variants in our dataset: *FLNC*, where heterozygous variants are already associated with adult-onset cardiomyopathy; and *GIGYF1* and *SYCP2*, which both include inframe indels that have been called but have an unusual balance between forward and reverse reads that makes us suspicious that they may actually be false positives. Unfortunately, we therefore do not believe we are able to identify any putatively novel

embryonic lethal genes from our data.

- Points from the manuscript:

o Abstract and discussion: The penetrance estimate of PZMs compared to DNMs needs to be addressed with a bit more care. The argumentation for this is not easy to follow. While this observation may be true for the overall list of PZMs; there will be a lot of PZMs that individually show 100% penetrance just like DNMs, so therefore this should be stated more carefully to avoid any misunderstanding.

We have slightly rephrased the abstract to make it clear that the reduced penetrance refers to the set of variants and not necessarily individual variants.

o Page 7, line 199: As part of the result section it would be good for the reader to understand that the 8464 DNMs are the exact same calls from the original DDD Nature manuscript (cite here accordingly).

We have added a comment and this reference.

o Page 7, line 194: describe 'clinical fit'. What is the expected outcome for a PZM in a gene for which germline DNMs cause severe DD – what about isolated ID.

We have expanded this point slightly in the text.

o Page 8, line 225: Of the 31 proven PZMs only 24 are considered (likely) pathogenic; although all candidate PZMs were considered likely pathogenic. It would be most interesting for the reader to understand the interpretation details for the remaining 7 variants. Did the VAF level influence this? Or what does this say about the initial criteria? Add also to Table 2.

We have added a comment about the 7 non-diagnostic PZMs in the manuscript, but have not added the variants to Table 2 as they are not diagnostic. The variants had a range of VAFs, which did not influence the decision.

o Page 9, line 288: It remains difficult to understand to what does 'these results' refer. The comparison of parental and patient PZMs may lead to too strong conclusions as these are two separate sets of mutations.

We have altered the paragraph structure to make it clear to which results we are referring.

* Parental-PZMs are mutations which in germline cause DD (as seen in the respective children); so yes indeed low level in blood of the parents does not cause an obvious phenotype. However were the parents really phenotyped in depths? No subtle phenotype? What about a low level mosaic in brain or other potentially affected tissues, was that excluded?

We have re-evaluated the phenotypic information we have for validated inherited parental mosaic variants, and can confirm that no phenotypes have been reported in the parents. However, the reviewer is correct that there may be a subtle unrecorded parental phenotype and the parents would be more correctly described as "apparently unaffected", which we have changed in the manuscript.

* Patient PZMs: TO a large extent the very same mutation was not observed in germline before. Hence viability and penetrance for those isn't fully known. AND: blood VAF may not say too much about e.g. brain VAF. In summary: The direct correlation between VAF and phenotype in blood seems oversimplified and needs to be discussed with greater care. Revertant mosaicism is not touched at all, which may be more common in blood.

We have removed the final concluding sentence from this section. Note that our analysis does not depend on any assumptions about blood/saliva VAF representing the level of mosaicism in other tissues.

o Page 11, line 336: Explain 'enrichment' in this context, this is enrichment over what?

We have rephrased this sentence.

o Page 12, line 387: The statement 'missing pathogenic mosaic variants from blood may be

considerably less important' is oversimplifying and may be misleading. This may give clinicians the wrong impression that a 2nd sample may not be valuable. There are ample examples that a 2nd (non-blood) sample is crucial e.g. Cornelia de Lange Syndrome, or any other 'organ-specific' or 'organ-restricted' phenotype. This needs to be re-written.

That is certainly not what we intended to imply, so we have rephrased this sentence.

Reviewer #4 (Remarks to the Author):

The authors have sufficiently addressed the concerns raised by Reviewer #1 in the previous round of review. The new analysis they have added, showing an intermediate methylation profile in a mosaic DDD proband, is quite interesting and helps improve the novelty of the study. Although this result only pertains to a single proband, I believe it is an important finding because it directly supports the intuition that mosaicism can lead to milder phenotypes in developmental disorders, as mentioned in lines 46 and 228-230.

Major comments:

1. The findings of the intermediate methylation profile in a mosaic DDD proband are currently buried in the Discussion (lines 344-352) and the methods leading to this result are only superficially mentioned in the caption of Supplementary Fig. 6. These should be moved to the Results and Methods, respectively, and the authors should explain in greater detail the data and methods used for this analysis.

We have made these additions and moved the Figure into the main text (Figure 5).

Minor comments:

2. The authors' justification for their use of different FDR thresholds is reasonable, and they appeal to the results shown in Fig. 4 to support their claim that there is no systematic difference in VAF between true mosaic benign mutations and true mosaic pathogenic mutations. This claim could be strengthened by testing the association of mosaic VAF in saliva and mosaic VAF in blood using linear regression and including DDG2P status as a covariate. If the authors' claim is correct, DDG2P status should not significantly improve the model fit. It would also be useful if Fig. 4 showed separate lines-of-best fit for the mosaic benign and mosaic pathogenic data.

We performed this analysis and confirmed that the association of VAF in blood vs saliva is not significantly different using DDG2P as a covariate. We have adjusted Figure 4 as suggested and added a comment to the figure legend.

Reviewer #5 (Remarks to the Author):

Review of ms 196604 Nat Comms entitled "Clinically-relevant postzygotic mosaicism in parents and children with developmental disorders in trio exome sequencing data" by Wright et al.

This manuscript describes an approach to analyse mosaic mutations (i.e. SNVs and indels –deviation from 50:50 for child-PZM and down to VAF levels ~1% for parent-PZM) in the large WES trio sequencing dataset generated by the DDD consortium (4293 trios).

Although the general approach is not novel and the results are in line with those described in previous studies, overall, I believe that this well-written manuscript represents a comprehensive and helpful addition to the recent body of work on mosaicism. The main attraction of the present study is the systematic approach for identification of mosaic SNVs (both parental- and child-PZM) across a large cohort – although with an average of 50X coverage for WES, the dataset is likely underpowered to identify mosaic cases (and overall only a small number of mosaic mutations are called).

I have a few comments and have identified some relatively minor issues that will need to be addressed:

1. Line 127 describes "a high-sensitivity set of 8,542 rare (MAF<0.01) candidate DNMs from 4,293

WES trios” but on line 199 (and line 160), the “high-sensitivity set of 8,464 candidate DNMs in 4,293 children with DDs” – please clarify.

The difference is due to the exclusion of X chromosome variants in males, due to the additional complexity of evaluating mosaicism for hemizygous variants. This has been clarified in the manuscript.

2. Despite applying a filtering threshold on proband of VAF>0.1 (line 136), some child-PZM were called and validated at VAF 0.04 (line 217 and Figure 3). This probably relates to Read-Depth but please explain or check phrasing?

The VAF>0.1 only applied to de novo calls, as stated in the methods.

3. Some comments on Table 2:

Please indicate the tissue that was used for the WES (relates to the “Mosaic VAF” column)

Please add the read depth for the WES VAF data (although read depth for the cohort is ~50X average, the likelihood to detect mosaic variants depends on local coverage).

The points above are important because the VAFs measured by WES data and the tissue deep-sequencing do not seem to always correlate – could you please plot and comment.

Please add the encoded protein change of the validated pathogenic changes

Patient 273554 with NFIX frameshift mutation – as annotated this patient is a Proband-PZM (VAF ~0.35 in both blood and saliva) but has an affected sib? This makes no sense?

We have made these changes to Table 2. The annotation of proband 273554 is correct – we assume the sib is either affected by a different condition, or that they share a partial dual diagnosis (excluding the mosaic variant in the proband).

4. I could not find a list of the 49 validated synonymous mutations (nor the other 12 validated mutations that did not make it on the Table 2 of pathogenic mutations) or whether validation was performed in blood and saliva (where available). These data would represent another estimate of the VAF correlation in these 2 tissues and with those measured by WES (I appreciate they are likely to have been plotted on Figure 4 but a supplementary table would be useful).

We have added a supplementary table of all validated mosaic variants.

5. It would be of interest to try to characterise the mutational signature of these mosaic mutations – although I suspect there are not enough calls to perform this analysis?

We agree but the reviewer is correct that there are too few variants for an informative analysis.

6. As mentioned in the text for pathogenic mutations, it is possible that there is a tissue-specific enrichment/depletion of mutant cells and therefore caution should be taken in interpreting the VAF levels measured in blood and/or saliva and those that may be present in other organs (including brain or gonads) and hence the correlation between observed VAF and phenotype or recurrence risk – it would be helpful to discuss this point in further details for example on lines 228-230 or 383-387. For some disorders such as Dravet or Cornelia de Lange, this is very relevant.

We have added some further comments into the text.

Related to this, 0.5% of parental-PZM were called – would you please comment on /discuss the benefit to assess germline tissue (i.e. sperm) to properly estimate paternal gonadal mosaicism – this matters also for interpretation of Fig 1B where “sibling recurrence risk” is correlated to low/high-level of Parent-PZM).

Several studies have now shown the power of sperm sequencing for direct recurrence risk estimation for paternally-derived DNMs (see for example Ref 8 - Breuss et al). Of note, this is particularly relevant for some DD genes such as SCN1A (see Yang et al Sci Rep 2017 - doi: 10.1038/s41598-017-15814-7) or MecP2 (doi: 10.1038/s41436-018-0348-2).

We have added a comment and references to the text.

7. Among all the validated mosaic (3% of causative child-PZM and the 0.5% parent-PZM), could you indicate how many (and which one, on Table 2 for example) had been previously identified (either as

DNM or mosaic) in previously published DDD studies. The abstract (line 31) and text (Line 275) state that together child-PZM and parent-PZM represent an additional 40 diagnoses. Is this so or is it a 'differential' diagnosis in some cases.

Most of the variants that were detected by our *de novo* mutation detection pipeline were previously published in a supplementary table in our Nature 2017 paper, though were neither identified as mosaic nor diagnostic in that paper; a small number of *de novo* variants and some of the inherited variants were published in our Genetics in Medicine 2018 paper as diagnoses, though not always as mosaic. We have identified these variants in Table 2 and have removed the word "additional" in the text.

Sup fig2 – correct “consitutive DNM in child”

Corrected.

Sup fig 3 – please indicate the number of trios for which avge mutations are plotted for each age group.

Replotted with size of point representing number of trios.

Reviewers' Comments:

Reviewer #2:

Remarks to the Author:

The authors have answered all requests satisfactory, and as such have further improved this manuscript. I can therefore recommend this manuscript for publication.

Reviewer #4:

Remarks to the Author:

The authors have sufficiently addressed all my previous comments.

Reviewer #5:

Remarks to the Author:

I have reviewed the changes in this latest version. The authors have clarified the remaining queries and have appropriately answered the reviewers' comments.

This is a well executed study that will be valuable to the field.

I have no further comments on this manuscript.